# What is a meaningful life for persons with acquired neurological impairments? A scoping review protocol

**Randi Steensgaard**[1,2]* , **Michele Offenbach Hundborg**[1,3], **Hanne Pallesen**[4,5],
**Lena Aadal**[4,5]

**1** Knowledge Centre for Neurorehabilitation of Western Denmark, Roskilde, Denmark, **2** Spinal Cord Injury Centre of Western Denmark, Neurology, Central Regional Hospital, Viborg, Denmark, **3** Specialised Center for Brain Injury, Central Denmark Region, Viborg, Denmark, **4** Hammel Neurorehabilitation Centre and University Research Clinic, RM, University of Aarhus, Aarhus, Denmark, **5** Department of Clinical Medicine, Aarhus University, Aarhus, Denmark

☯ These authors contributed equally to this work.
* rst@specialhospitalet.dk

**Data Availability Statement:** No datasets were generated or analysed during the current study. All relevant data from this study will be made available upon study completion.

## Abstract

### Objective

This scoping review explores the constitution of a meaningful life as perceived by adults with acquired neurological impairment following an injury or a disease.

### Introduction

A neurological injury or disease imposes extensive life changes on the affected person and his or her close relatives. Including the patients' perception of a meaningful life is crucial to facilitate adjustment of any rehabilitation initiatives to the patients' wishes, hopes, needs, and preferences. Even so, the descriptions and common traits of a meaningful life from the impaired person's perspective are scarcely covered in the literature. Hence, a scoping review of existing knowledge is needed to facilitate quality rehabilitation and research initiatives.

### Inclusion criteria

All studies, regardless of their design, are included provided they describe a meaningful life as considered or experienced by persons aged 18 years or more with neurological impairment.

### Methods

A PICo framework defines the search algorithms used in the databases MEDLINE, Cinahl, PsycINFO and Embase. Using Covidence, the scoping review systematically organizes the identified articles to provide a broad description of the study phenomenon. Furthermore, titles, abstracts, and full-text articles are screened independently by two reviewers to determine if they meet the inclusion criteria. In case of disagreement, a third and fourth reviewer are consulted. The scoping will be reported according to the PRISMA- SCR checklist.

**Funding:** The authors received no specific funding for this work.

**Competing interests:** The authors have declared that no competing interests exist.

## Introduction

A neurological disease or injury imposes extensive life changes on the impaired person and his/her close relatives [1–3]. Neurological impairment covers a wide range of illnesses and injuries. Depending on the severity and type of impairment the person can be more or less affected. For instance, adults with acquired brain injury (ABI) often experience changes in sensory-motor, cognitive and psychological functions affecting the quality and meaningfulness of life [4]. For persons with a spinal cord injury cognitive functions are maintained. Nevertheless, this group may suffer from tetraplegia meaning that they are unable to use both arms and legs sufficiently leaving them highly dependent on others. Multiple Sclerosis, being a progressive illness, affects the person's function and abilities in a descending spiral. Common for all there are wide range of negative and severe psycho-social consequences, including a low quality of life, low self-determination, low participation and autonomy, low attachment to the labor market, and risk of stigmatizing [5–11]. Furthermore, physical impairment and reduced physical mobility and activity are additional well-known consequences that may inhibit the return to former everyday activities [12].

Rehabilitation may help a person to adjust and redefine everyday activities following neurological impairment [13]. To underpin the patients' motivation to fully engage in highly intensive rehabilitation programs, it is essential to establish what the affected person finds important [14]. Hence, including patients' wishes, hopes, and preference, and adjusting activities to new realities are of paramount importance to the rehabilitation process [15,16] and central to successful rehabilitation [17,18].

According to the World Health Organization (WHO), rehabilitation is defined as "(. . .) appropriate measures, including through peer support, to enable persons with disabilities to attain and maintain their maximum independence, full physical, mental, social and vocational ability, and full inclusion and participation in all aspects of life" [19,20].

With this definition, the WHO does not directly address a meaningful life. They do, however, highlight the need to attain full inclusion and participation in all aspects of life. Meyer et al. (2020) [21] described the problematic lack of a unified definition of rehabilitation, specifying that the aim of rehabilitation is the optimization of aspects of functioning, especially social participation, independence or self-determination, or quality of life as experienced by the individual. The aspect of a personal, perceived, and experienced measure is even more explicit in Denmark where a meaningful life is described as the end point of the rehabilitation process [22–24]. Moreover, a meaningful life remains the ultimate goal of rehabilitation according to the second version of the Danish White Book on rehabilitation, which will be published in February 2022.

Several studies have indicated that person-centered rehabilitation and increased focus on the patient's perception of meaningful life improve the outcomes for the individual, including their quality of life [25–29]. Furthermore, such a focus may encourage persons to engage more fully in their rehabilitation process, leading to individual support and efforts being adjusted further to the individual's wishes, hopes, and needs [26]).

Despite the positive outcome achieved by including the individual's preferences, little is known about which aspects and elements are commonly perceived as meaningful by persons with neurological impairments. The content of the term 'meaningful life' is often used in different contexts [22,23]. However, there is neither consensus about the content of a meaningful life nor a clear definition. A preliminary search for existing scoping reviews and systematic reviews on the topic has been conducted in MEDLINE and the Cochrane Database of Systematic Reviews. No relevant systematic or scoping reviews were found. Hence a scoping review is warranted to summarize and describe what individuals with neurological impairment experience as a meaningful life.

## Aim

To identify how a meaningful life is perceived by adults with acquired neurological impairments.

## Materials and methods

The scoping review will be conducted in accordance with the Joanna Briggs Institute methodology for scoping reviews [30]. Further, the scoping review will be reported using the scoping review PRISMA-SCR checklist. The checklist contains 20 essential reporting items and 2 optional items to include when completing a scoping review [31].

An interdisciplinary team of researchers within the field of neurological rehabilitation will systematically identify, retrieve, review, and synthesize international evidence of relevance to elaborate on the phenomenon of meaningful life with a neurological impairment.

### Inclusion criteria

**Participants.**   Data are included when reflecting first-hand experiences of persons > 18 years with an acquired neurological impairment. Children may have another lived experience and understanding of a meaningful life. Furthermore, they are under the influence of guardians and their perception of a meaningful life. Therefore, studies with children are not included. Nevertheless, it may not be possible to detect those who acquired their impairment under the age of 18 if this is not stated in the studies. In this case, they are included with their adult perspective. Studies are included when they provide perspectives from patients with progressive conditions such as e.g. multiple sclerosis and patients with acquired impairment (e.g. stroke, SCI, TBI).

### Context

This review is not limited to a particular country or healthcare system. The review will consider studies conducted in all settings (e.g. rehabilitation centers, primary care, long-term care institutions, at home, in the municipalities, or other care facilities).

Studies published in English, Danish, Swedish, and Norwegian will be included. Studies are not limited by publication year.

**Information sources.**   All studies, regardless of their design (quantitative, qualitative, or mixed-methods studies) and quality, are included provided they describe a meaningful life as considered or experienced by patients and persons with neurological impairments. In addition, systematic reviews that meet the inclusion criteria are considered, depending on the research question. Grey literature including reports, expert opinions and editorials are also considered for inclusion in this scoping review.

### Search strategy

The search strategy is developed in collaboration with a research librarian. The first step will include an initial limited search of MEDLINE, CINAHL, PsycINFO, and Embase to identify articles on the topic. The text words contained in the titles and abstracts of relevant articles, and the index terms used to develop a PICo (**P**opulation, Phenomenon of **I**nterest, **Co**ntext) [32] to describe the articles and to develop a full search strategy (see S1 Appendix). The Population (P) is "Patients with a neurological injury or illness. The phenomenon of interest (I) is meaningful life and the Context (Co) is rehabilitation. The search strategy, including all identified keywords and index terms, is adapted for each of the included database MEDLINE, CINAHL, PsycINFO, and Embase and/or information source. The third step is to use the

reference list of all included sources of evidence is screened for additional studies (S1 Appendix shows an example of the search).

### Exclusion criteria

Studies will be excluded if they describe a meaningful life from the perspective of children younger than 18 years, persons who have another illness than neurological impairment or studies published in other languages than English or Scandinavian.

### Data extraction process and critical appraisal

Following the search, all identified citations will be collected and uploaded into EndNote (X9.2/ 2020) and duplicates will be removed. Any potentially relevant sources will be retrieved in full and their citation details will be imported into Covidence [33] for systematic review management. Each step of the review process will be conducted independently by two reviewers. In case of disagreement, consensus is established by discussion between the two reviewers. Alternatively, a third and fourth reviewer are consulted. Firstly, titles and abstracts will be assessed to determine if the studies comply with the inclusion criteria. Secondly, the full text of selected citations is assessed in detail against the inclusion criteria. Reasons for exclusion of full text sources of evidence that do not meet the inclusion criteria are recorded and reported in the scoping review.

Included studies are evaluated using a design specific quality assessment template i.e. the appropriate CASP checklist [34]. Furthermore, a data extraction template developed and pilot tested by the reviewers will contribute to ensuring a systematic data collection and analysis.

The results of the search and the study in- and exclusion process will be reported in full with reasons for exclusions at each stage in the final scoping review and presented in a PRISMA-ScR flow diagram [31].

### Data analysis and presentation

Using Covidence [33], data extraction will involve all authors using a draft data extraction tool. Independent extraction will be the following data from the articles: citations (authors, title, publication year, and journal), country (of origin), sample size, study design, population, study aim, and extracted relevant data. Data analysis will be conducted at physical meetings between all authors. The extracted data will be presented in a diagrammatic or tabular form in a manner that aligns with the objective of this scoping review. An inductive narrative summary will accompany the tabulated and/or charted results, which will describe how the results relate to the review's question.

### Ethical considerations and declarations

This study will not include participants and will be conducted in accordance with the Helsinki II Declaration and Ethical Guidelines for Nursing Research in the Nordic Countries.

### Study status and timeline

Fig 1 shows the progress and of the scoping review.

## Discussion

This scoping review will provide knowledge on the patient perspective on meaningful aspects of life. This is essential when health professionals wish to "enable persons with disabilities to attain and maintain their maximum independence, full physical, mental, social and vocational ability, and full inclusion and participation in all aspects of life" as stated by WHO [20]. With the knowledge of this scoping review rehabilitation can become more qualified and targeted the individual

| 2021 | |
|---|---|
| **August – September** | **Data extraction** <br> **Data assessment** |
| **October - December** | **Analysis** |
| **2022** | |
| **January** | **Write and publish review** |

**Fig 1. Timeline of the development of the scoping review.**

person's life. Accordingly, health professionals will have a clearer idea of important aspects for person to participate in and what elements of inclusion that may be meaningful to the individual.

Using the broad term "neurological impairment" we cluster persons with different illnesses and diseases. This can be necessary in order to get rigor and enough articles to make a careful analysis. However, we also risk to compare persons with very different disabilities. We hope this scoping review will initiate more research within the neurological field on meaningful life.

The scoping review report will be published in a peer-review journal. The results will be disseminated at rehabilitation conferences in Denmark and internationally. Furthermore, the results will consolidate the work of the West Danish Knowledge Centre on Neuro Rehabilitation and the ongoing development of rehabilitation in Denmark.

## Supporting information

**S1 Appendix. Search strategy.**
(DOCX)

## Acknowledgments

The authors hereby acknowledge the meticulous support and guidance provided by the librarians at the Central Regional Hospital during our literature search.

## Author Contributions

**Conceptualization:** Randi Steensgaard, Michele Offenbach Hundborg, Hanne Pallesen, Lena Aadal.

**Data curation:** Randi Steensgaard, Michele Offenbach Hundborg, Hanne Pallesen, Lena Aadal.

**Formal analysis:** Randi Steensgaard, Michele Offenbach Hundborg, Hanne Pallesen, Lena Aadal.

**Investigation:** Randi Steensgaard, Michele Offenbach Hundborg, Hanne Pallesen, Lena Aadal.

**Methodology:** Randi Steensgaard, Michele Offenbach Hundborg, Hanne Pallesen, Lena Aadal.

**Writing – original draft:** Randi Steensgaard, Michele Offenbach Hundborg, Hanne Pallesen, Lena Aadal.

**Writing – review & editing:** Randi Steensgaard, Michele Offenbach Hundborg, Hanne Pallesen, Lena Aadal.

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
