## [Decision Letter · Decision Letter 0]

24 Nov 2021

PONE-D-21-35911What is a meaningful life for persons with acquired neurological impairments? A scoping review protocolPLOS ONE

Dear Dr. Steensgaard,

Thank you for submitting your manuscript to PLOS ONE. After careful consideration, we feel that it has merit but does not fully meet PLOS ONE’s publication criteria as it currently stands. Therefore, we invite you to submit a revised version of the manuscript that addresses the points raised during the review process. Both reviewers indicated the merit of your review but raised a number of issues regarding your methodological approach.  Specifically, they both indicated the need for more details regarding your intended methodology for your scoping review, which includes providing details on which scoping review framework you intend to follow, your inclusion/exclusion criteria, and how your approach will adhere to the PRISMA-SCR. I have reviewed your protocol as well and agree with both reviewers that there are significant details lacking in the protocol as written.  I would also like to see more background information on the types of neurological impairments that might be included in your review and a more nuanced discussion of the impact of these impairments on the person's health and wellbeing. The discussion is also lacking some discussion on the larger impact of this review to the field aside from being published in a journal and helping the field in the local context.

We look forward to receiving your revised manuscript.

Kind regards,

Sander L. Hitzig

Academic Editor

PLOS ONE

2.We note that you have stated that you will provide repository information for your data at acceptance. Should your manuscript be accepted for publication, we will hold it until you provide the relevant accession numbers or DOIs necessary to access your data. If you wish to make changes to your Data Availability statement, please describe these changes in your cover letter and we will update your Data Availability statement to reflect the information you provide.

Reviewers' comments:

Reviewer's Responses to Questions

**Comments to the Author**

1. Does the manuscript provide a valid rationale for the proposed study, with clearly identified and justified research questions?

Reviewer #1: Yes

Reviewer #2: Yes

2. Is the protocol technically sound and planned in a manner that will lead to a meaningful outcome and allow testing the stated hypotheses?

Reviewer #1: Partly

Reviewer #2: Partly

3. Is the methodology feasible and described in sufficient detail to allow the work to be replicable?

Reviewer #1: No

Reviewer #2: No

4. Have the authors described where all data underlying the findings will be made available when the study is complete?

Reviewer #1: Yes

Reviewer #2: Yes

5. Is the manuscript presented in an intelligible fashion and written in standard English?

Reviewer #1: Yes

Reviewer #2: Yes

6. Review Comments to the Author

You may also provide optional suggestions and comments to authors that they might find helpful in planning their study.

Reviewer #1: Hello,

Thank you for the opportunity to review this interesting work. However, much more detail is required.

- How is neurological conditions being defined? Is this for all age groups?

- How many reviewers will partake on each step of the methodology? How will screening occur.

- Can you better define "grey literature"? Specifically, what methods did you use to explore the grey literature and what inclusion/exclusion you used to initially map out all available "grey literature"?

-Can you better define what you mean by "meaningful life"?

- What scoping review methodology are you following and how is the PRISMA-SCR being adhered to?

-The Keywords are clear but for indexing questions I suggest that the terms should be extracted from MeSH.

-It is necessary to justify the choice of this time limit (or lack thereof)

-It is necessary to justify the inclusion of all study designs (e.g., why are other review papers going to be included and how will they be synthesized)

-Despite being a literature review there is ethical issues associated with this type of research that should be reported- what does this look like in terms of the "the Helsinki II Declaration and Ethical Guidelines for Nursing Research in the Nordic Countries" cited

- What studies will be excluded and why?

-The team present the categorization of the content of the articles from the bibliographic sample (i.e., through Covidence) - please clarify the methodological procedures and how this will be reported. Will results of the studies be included? How will themes be developed? Refer to the PRISMA-SCR item 13: Detail how evidence will be presented which may be in a

narrative format, table, or visual representation, including a map or diagram

- Please justify the use of the CASP in a scoping review (that typically does not assess methodological quality), why was this chosen? Will this apply to all study designs?

Reviewer #2: This scoping review protocol paper highlights the importance of understanding how a meaningful life is perceived by adults with acquired neurological impairments. Indeed, this would be an important review to add to the already existing literature. However, I have several issues with the methods used by the authors that need to be addressed should you want this protocol to be published.

Introduction

1. Line 55 - Spell out acronyms the first time they appear (World Health Organization – WHO).

2. Good rationale for choosing to understand meaningful life in persons with neurological impairments – I would agree that participants thoughts about what they believe are meaningful to their quality of life are not often discussed.

3. The last sentence of the introduction could be more clear/concise. I would consider removing “what is known about the content of the ultimate goal of rehabilitation” since you are focusing on a meaningful life in the context of rehabilitation.

Materials and Methods

1. What scoping review framework are you following for this review? This is a critical piece that is missing from this protocol. There are established frameworks that are widely used and are the gold standards for conducting scoping reviews. (Levac et al., 2010, - https://pubmed.ncbi.nlm.nih.gov/20854677/ and the Joanna Briggs Institute - https://journals.lww.com/ijebh/Fulltext/2015/09000/Guidance_for_conducting_systematic_scoping_reviews.5.aspx).

2. It is a requirement of this journal that the appropriate PRISMA checklist is used for submission of scoping review protocols. Please mention that you are following the PRISMA guidelines and consider following their items for reporting your methods for this review. This will help ensure that your review is transparent and reproducible which is pivotal for this type of work.

3. Eligibility Criteria – this needs to be further fleshed out.

a. Why are you focusing on those >18 years or older? Are you only including individuals who had their injury after that age?

b. How are you conceptualizing meaningful life? What must articles discuss to be considered as “describing a meaningful life”?

c. I am confused about line 89. Are you only including studies if they collected data in person? Does this exclude telephone

surveys or online interviews? Please be more explicit by what you mean by this.

4. Information sources – I am pleased to see that you are including grey literature as this is becoming more standardized for scoping reviews. However, not all grey literature is created equal (i.e., opinion pieces are not generally accepted). Please be more explicit with the grey literature you are choosing to use, how you will be searching for it and what sources you will use to search for the appropriate grey literature.

5. Search Strategy – You say that an initial limited search will be done on 4 databases but the example search strategy you provided is for PubMed. Do you intend to search PubMed as one of your databases? PubMed is generally not recommended for scoping reviews because it is hard to replicate the search results, so I would suggest sticking with the initial databases you mentioned. Furthermore, you need to be more explicit about what databases you are using in your final search.

6. Data extraction process and critical appraisal:

a. Will you be doing a pilot to ensure that the two independent reviewers understand the inclusion criteria? This is

recommended in the frameworks (Levac, JBI) for conducting rigorous scoping reviews. If you are doing a pilot, will the

remaining articles be split or will both reviewers review all the articles for inclusion.

b. I appreciate that you are doing a quality assessment. I would suggest saying that you will be using the appropriate CASP

checklist because there are multiple quality assessment templates that have been put out by them.

7. Data analysis and presentation – the information you have presented here should actually be in the data extraction process. In the analysis section you should describe what techniques you will be using to summarize/collate the data, not what information you will be taking from the articles.

Discussion

1. How could the information you obtain from the review be used by those working in rehabilitation? I think this is a major point missing from the discussion.

7. PLOS authors have the option to publish the peer review history of their article (what does this mean?). If published, this will include your full peer review and any attached files.

Reviewer #1: No

Reviewer #2: No

---

## [Author Response · Author response to Decision Letter 0]

21 Dec 2021

Response to editor and reviewers

Editor:

More details regarding your intended methodology for your scoping review

- details on which scoping review framework

- your inclusion/exclusion criteria

- how adhere to the PRISMA-SCR

Answer: 

Joanna Briggs – we have added this

We have elaborated on this.

We have used the checklist in the document with track chances to illustrate how we plan to describe this

More background information on the types of neurological impairments that might be included

Answer:

We have added text in the introduction (line 45-52)

A more nuanced discussion of the impact of these impairments on the person's health and wellbeing

Answer:

We have added text in the introduction and discussion.

Discussion on the larger impact of this review

Answer:

A good and important point – we have added that in the discussion section (line 159-173)

Protocol should be reported in sufficient detail for another researcher to reproduce all experiments and analyses

Answer:

We have described our steps more clearly

Reviewer 1:

- How is neurological conditions being defined? Is this for all age groups?

Answer:

We wish to be open to all neurological conditions. In our search we searched for:

("Nervous System Diseases"[MeSH Terms] OR ("brain injur*"[Text Word] OR "spinal cord trauma*"[Text Word] OR "spinal cord injur*"[Text Word] OR "spinal cord transection*"[Text Word] OR "spinal cord lacer*"[Text Word] OR "traumatic myelopath*"[Text Word] OR "spinal cord contusion*"[Text Word] OR "stroke*"[Text Word] OR "apoplex*"[Text Word] OR "cerebrovascular accident*"[Text Word] OR "brain vascular accident*"[Text Word] OR "multiple sclerosis"[Text Word] OR "disseminated sclerosis"[Text Word] OR "brain laceration*"[Text Word] OR "parkinson*"[Text Word] OR "paralysis agitans"[Text Word] OR "amyotrophic lateral sclerosis"[Text Word] OR "lou gehrig disease*"[Text Word]))

We did limit the search for persons older than 18 years.

- How many reviewers will partake on each step of the methodology? How will screening occur.

Answer:

We have clarified this (line 131-133). We are at least two independent reviewers conducting each step of the scoping review.

- Can you better define "grey literature"? Specifically, what methods did you use to explore the grey literature and what inclusion/exclusion you used to initially map out all available "grey literature"?

Answer:

We have defined grey literature (line 112-113)

-Can you better define what you mean by "meaningful life"?

Answer:

The aim of the study is to describe what a meaningful life is from a patient. perspective. This requires us to stay open to whatever the persons characterise as meaningful. Therefore, we have not made any prior statements or definitions on this topic.

- What scoping review methodology are you following and how is the PRISMA-SCR being adhered to?

Answer:

We follow the Joanna Briggs Institute methodology for scoping reviews and have added PRISMA-SCR. 

Throughout the entire document the track changes now provide information on how the PRISMA-SCR is being adhered.

-The Keywords are clear but for indexing questions I suggest that the terms should be extracted from MeSH.

Answer:

We absolutely agree that this is a good idea. Neurology and rehabilitation are MESH terms.

Scoping review is not a MESH term. However, we will keep this keyword as it is more correct than those suggested in the MESH database. The same goes for meaningful life.

-It is necessary to justify the choice of this time limit (or lack thereof)

Answer:

We only present a timeline for the work and we are uncertain about what we need to elaborate on concerning time limit.

-It is necessary to justify the inclusion of all study designs (e.g., why are other review papers going to be included and how will they be synthesized)

Answer:

We include all designs in order to gain the best possible illumination of the topic. Inclusion of reviews are a widely discussed issue. We are aware of the interpretation and will use the reference list for both critical evaluations of the statements and line search.

-Despite being a literature review there is ethical issues associated with this type of research that should be reported- what does this look like in terms of the "the Helsinki II Declaration and Ethical Guidelines for Nursing Research in the Nordic Countries" cited

Answer:

Ethical considerations are described (line 175 – 178)

The fact that we do not include participants directly there are some ethical aspects we donot have to account for. However, according to the Ethical guidelines for Nursing research in Nordic countiries (https://ssn-norden.dk/wp-content/uploads/2020/05/ssns_etiske_retningslinjer_0-003.pdf) 

the researcher has a duty to comply with the ethical guidelines laid down in the legislation, conventions or declarations which apply to all researchers. This duty covers all phases of the research process. The researcher should have good knowledge of methods and should possess the competence required by the project, both with regard to subject and to method. This means that the researcher must be qualified for and have experience in applying relevant methods for the collection and processing of data. Researchers who do not have research experience must work under the supervision of researchers with experience. The researcher shall ensure the safe storage of research materials. The researcher shall comply with prevailing national and international rules for authorship. Co-authorship entails that a substantial scientific contribution is made which comprises formulating the project or project-specific methods, or making an analysis and a critical interpretation of data, as well as participating in the formulation or critical examination of the manuscript. Further, it is the duty of the researcher to make all results publicly available and to publish them – including possible negative results. 

We adhere to these aspects of the guideline. The same considerations are made about the Helsinki declaration where we adhere to the Scientific Requirements and Research Protocols.

(https://www.wma.net/policies-post/wma-declaration-of-helsinki-ethical-principles-for-medical-research-involving-human-subjects/ )

- What studies will be excluded and why

Answer: 

We have added this (line 135-136, 140-141).

-The team present the categorization of the content of the articles from the bibliographic sample (i.e., through Covidence) - please clarify the methodological procedures and how this will be reported. Will results of the studies be included? How will themes be developed?

Answer:

We have elaborated the methodological procedures (line 143-149)

Refer to the PRISMA-SCR item 13: Detail how evidence will be presented which may be in a

narrative format, table, or visual representation, including a map or diagram

Answer:

This is added.

- Please justify the use of the CASP in a scoping review (that typically does not assess methodological quality), why was this chosen? Will this apply to all study designs?

Answer:

Thank you for being critical. Included studies are evaluated using a design specific quality assessment template i.e. the CASP checklist for qualitative studies (line 137-139). The quality assessment is needed, as we have no time limit for inclusion and the research question will predominantly be elucidated by qualitative studies where the quality has evolved over decades. 

Reviewer 2

Introduction

1. Line 55 - Spell out acronyms the first time they appear (World Health Organization – WHO).

Answer:

We have added this

2. Good rationale for choosing to understand meaningful life in persons with neurological impairments – I would agree that participants thoughts about what they believe are meaningful to their quality of life are not often discussed.

Answer:

Thank you for agreeing on this.

3. The last sentence of the introduction could be more clear/concise. I would consider removing “what is known about the content of the ultimate goal of rehabilitation” since you are focusing on a meaningful life in the context of rehabilitation.

Answer:

We have made the recommended change.

1. What scoping review framework are you following for this review? This is a critical piece that is missing from this protocol. There are established frameworks that are widely used and are the gold standards for conducting scoping reviews. (Levac et al., 2010, - https://pubmed.ncbi.nlm.nih.gov/20854677/ and the Joanna Briggs Institute

Answer:

Joanna Briggs Institute methodology for scoping reviews. We have added this.

Throughout the entire document with track changes we have now provided information on how the PRISMA-SCR is being adhered to.

2. It is a requirement of this journal that the appropriate PRISMA checklist is used for submission of scoping review protocols. Please mention that you are following the PRISMA guidelines and consider following their items for reporting your methods for this review. This will help ensure that your review is transparent and reproducible which is pivotal for this type of work.

Answer: 

Thank you for this comment. We have now added this.

3. Eligibility Criteria – this needs to be further fleshed out.

a. Why are you focusing on those >18 years or older? Are you only including individuals who had their injury after that age?

b. How are you conceptualizing meaningful life? What must articles discuss to be considered as “describing a meaningful life”?

c. I am confused about line 89. Are you only including studies if they collected data in person? Does this exclude telephone

surveys or online interviews? Please be more explicit by what you mean by this.

Answer:

In and exclusion criteria has been fleshed out (line 97-113, 124-127)

a) Our aim is to understand the overall aim of individualized rehabilitation "a meaningful life" from a first-hand perspective. Therefore, we include autonomous persons suffering from injuries acquired as adults.

b) The aim of the study is to describe what a meaningful life is from a patient. perspective. This requires us to stay open to whatever the persons characterise as meaningful. Therefore, we have not made any prior statements or definitions on this topic.

Meaningful life must be expressed of the patients or cited in the included articles. 

c) Sorry for that confusion. What we mean is that it should be a first-hand perspective. We hope this is clear. We include all methods – including telephone surveys if the questions are open-ended.

4. Information sources – I am pleased to see that you are including grey literature as this is becoming more standardized for scoping reviews. However, not all grey literature is created equal (i.e., opinion pieces are not generally accepted). Please be more explicit with the grey literature you are choosing to use, how you will be searching for it and what sources you will use to search for the appropriate grey literature.

Answer:

Grey literature including reports, expert opinions and editorials are also considered for inclusion in this scoping review (112-113). Initiating search using Google, Google Scholar and the ScoPus database which also contains references to conference proceedings and websites will be used. (not described in article due to word limit)

5. Search Strategy – You say that an initial limited search will be done on 4 databases but the example search strategy you provided is for PubMed. Do you intend to search PubMed as one of your databases? PubMed is generally not recommended for scoping reviews because it is hard to replicate the search results, so I would suggest sticking with the initial databases you mentioned. Furthermore, you need to be more explicit about what databases you are using in your final search.

Answer:

Medline (PubMed) was recommended by the Liberian in order to identify relevant text and keyword to develop the systematic search strategy.

Databases used (line 121-122). The search we have included in the article is just an example and we can include some of our additional searches as well.

6. Data extraction process and critical appraisal:

a. Will you be doing a pilot to ensure that the two independent reviewers understand the inclusion criteria? This is

recommended in the frameworks (Levac, JBI) for conducting rigorous scoping reviews. If you are doing a pilot, will the

remaining articles be split or will both reviewers review all the articles for inclusion.

b. I appreciate that you are doing a quality assessment. I would suggest saying that you will be using the appropriate CASP

checklist because there are multiple quality assessment templates that have been put out by them.

Answer:

(a) We have elaborated the methodological procedures (line 129-141, 143-149)

(b) Thank you – please see line (137-138)

7. Data analysis and presentation – the information you have presented here should actually be in the data extraction process. In the analysis section you should describe what techniques you will be using to summarize/collate the data, not what information you will be taking from the articles.

Answer:

We agree and have elaborated in line 143-149

Discussion

1. How could the information you obtain from the review be used by those working in rehabilitation? I think this is a major point missing from the discussion

Answer:

We agree. Thanks for pointing this out. We have added a paragraph concerning this.

---

## [Decision Letter · Decision Letter 1]

16 May 2022

What is a meaningful life for persons with acquired neurological impairments? A scoping review protocol

PONE-D-21-35911R1

Dear Dr. Steensgaard,

We’re pleased to inform you that your manuscript has been judged scientifically suitable for publication and will be formally accepted for publication once it meets all outstanding technical requirements.

Kind regards,

Avanti Dey, PhD

Staff Editor

PLOS ONE

Additional Editor Comments (optional):

Thank you for thoroughly addressing the reviewers' comments. Please note that Reviewer #2 has noted a couple minor suggestions which we kindly ask that you address in your final version. 

Reviewers' comments:

Reviewer's Responses to Questions

**Comments to the Author**

1. Does the manuscript provide a valid rationale for the proposed study, with clearly identified and justified research questions?

Reviewer #2: Yes

2. Is the protocol technically sound and planned in a manner that will lead to a meaningful outcome and allow testing the stated hypotheses?

Reviewer #2: Yes

3. Is the methodology feasible and described in sufficient detail to allow the work to be replicable?

Reviewer #2: Yes

4. Have the authors described where all data underlying the findings will be made available when the study is complete?

Reviewer #2: Yes

5. Is the manuscript presented in an intelligible fashion and written in standard English?

Reviewer #2: Yes

6. Review Comments to the Author

You may also provide optional suggestions and comments to authors that they might find helpful in planning their study.

Reviewer #2: Thank you for addressing my comments. Overall, you have done a great job with my feedback and I am pleased with the responses given and the updates that have been added to the manuscript. I have two minor comments for consideration:

1. Very minor but Medline is not a PubMed database, it is part of OVID. I appreciate that the librarian has recommended Medline and not Pubmed.

2. In your inclusion/exclusion criteria, you don't mention why you're choosing to exclude persons under 18. Furthermore, I would elaborate whether you are including people who received their injury prior to 18 years old and then provide their perspective as adults or whether you only want persons who had their impairment after they turned 18. Perspectives may be different for both groups so important to flesh out.

7. PLOS authors have the option to publish the peer review history of their article (what does this mean?). If published, this will include your full peer review and any attached files.

Reviewer #2: No

---

## [Editor Report · Acceptance letter]

8 Jun 2022

PONE-D-21-35911R1 

What is a meaningful life for persons with acquired neurological impairments? A scoping review protocol 

Dear Dr. Steensgaard:

I'm pleased to inform you that your manuscript has been deemed suitable for publication in PLOS ONE. Congratulations! Your manuscript is now with our production department. 

Kind regards, 

on behalf of

Dr. Avanti Dey 

Staff Editor

PLOS ONE